# Prevalence of Mental Health and Social Connection among Patients Seeking Tobacco Dependence Management: A Pilot Study

**DOI:** 10.3390/ijerph191811755

**Published:** 2022-09-17

**Authors:** Panagis Galiatsatos, MopeninuJesu Oluyinka, Jihyun Min, Raiza Schreiber, Dina G. Lansey, Ruth Ikpe, Manuel C. Pacheco, Victoria DeJaco, Alejandra Ellison-Barnes, Enid Neptune, Norma F. Kanarek, Thomas K. M. Cudjoe

**Affiliations:** 1Division of Pulmonary and Critical Care Medicine, Johns Hopkins School of Medicine, Baltimore, MD 21224, USA; 2The Tobacco Treatment and Cancer Screening Clinic, Baltimore, MD 21224, USA; 3Medicine for the Greater Good, Johns Hopkins School of Medicine, Baltimore, MD 21224, USA; 4Department of Oncology, Johns Hopkins School of Medicine, Baltimore, MD 21224, USA; 5Univeridad Tecnológica de Pereira, Universidad Visión de las Americas, Pereira 660003, Colombia; 6Environmental Health and Engineering, Johns Hopkins School of Public Health, Baltimore, MD 21224, USA; 7Division of Geriatric Medicine and Gerontology, Johns Hopkins School of Medicine, Baltimore, MD 21224, USA

**Keywords:** tobacco dependence, mental health, smoking cessation, social isolation

## Abstract

Introduction: with regards to tobacco dependence management, there are certain barriers to successful smoking cessation for patients, such as untreated anxiety and depression. Complicating the impact of mental health morbidities on tobacco dependence may be the significant portion of patients whose mental health issues and limited social connections are undiagnosed and unaddressed. We hypothesize that patients with no prior mental health diagnoses who are treated for tobacco dependence have high rates of undiagnosed mental health morbidities. Methods: patients were recruited from a tobacco treatment clinic in 2021. Every patient who came for an inaugural visit without a prior diagnosis of mental health disease was screened for depression, anxiety, social isolation and loneliness. Sociodemographic variables were collected. Results: over a 12-month period, 114 patients were seen at the tobacco treatment clinic. Of these 114 patients, 77 (67.5%) did not have a prior diagnosis of a mental health disease. The mean age was 54.3 ± 11.2 years, 52 (67.5%) were females, and 64 (83.1%) were Black/African American. The mean age of starting smoking was 19.3 ± 5.2 years, and 43 (55.8%) had never attempted to quit smoking in the past. With regards to mental health screening, 32 (41.6%) patients had a score of 9 or greater on the Patient Health Questionnaire (PHQ) 9, 59 (76.6%) had a score of 7 or greater on the Generalized Anxiety Disorder (GAD) 7, 67 (87.0%) were identified with social isolation and 70 (90.1%) for loneliness on screening. Conclusion: there was a high prevalence of undiagnosed mental health morbidities and social disconnection in patients who were actively smoking and were struggling to achieve smoking cessation. While a larger scale study is necessary to reaffirm these results, screening for mental health morbidities and social disconnection may be warranted in order to provide effective tobacco dependence management.

## 1. Introduction

Tobacco smoking continues to be the leading risk factor in preventable morbidity and mortality in the United States (US) as well as resulting in significant healthcare-related economic expenditures [1,2]. For successful tobacco dependence management, in an effort to prevent or manage such dire health consequences, smoking cessation is vital. Since many patients do want to stop smoking altogether, clarifying the impact smoking is having on their health and well-being is key [3]. Over the last few decades, advancements in behavioral counseling and pharmacotherapies, along with the reach of such smoking cessation interventions, has resulted in more patients able to achieve smoking cessation, contributing to the current active smoking rate in the US at its lowest point recorded [4]. However, there are still patients who are refractory with regards to smoking cessation and, ultimately, tobacco independence. For such patients, an understanding of their barriers is key in an effort to create a more specific, and operational, tobacco dependence phenotype that would facilitate more precise interventions for management.

People with mental health morbidities, such as depression, anxiety and schizophrenia, are more prone to tobacco use and tobacco dependence due to their susceptibility for nicotine addiction [5,6,7]. A plausible explanation to this from a biological standpoint may have to do with the recognition that acetylcholine receptors in the mesolimbic region of the brain are involved in the pathogenesis for both mental diseases and nicotine addiction [8,9,10,11], enabling nicotine introduced by tobacco consumption to ameliorate some of the symptoms that patients experience from their mental health disorders [5,8,12]. While clinicians can assist persons who smoke by consistently discussing the concern and providing guidance, if there is an omission in understanding how and why the person smokes, often referred to as a patient’s smoking topography [13], this may result in an inability for a patient to manage their nicotine cravings during the time they are amplified by mental health issues (e.g., stress, anxiety) or social disconnection (e.g., isolation or loneliness).

Persons with mental illnesses are associated with heavier smoking of combustible cigarettes, greater dependence on nicotine, and lower smoking cessation attempts and successes [14,15,16,17]. Therefore, with such a severe tobacco dependence, it is not surprising that the prevalence of smoking in patients with mental health morbidities is estimated to be as high as 50% in the United States [1,5]. More concerning is that this prevalence of smoking may be greatly underreported given that mental health diseases are often underdiagnosed [18,19,20,21]. Since patients with mental health diagnoses are often highly motivated to quit smoking [22] and can achieve smoking cessation successfully through the dual management of their mental health morbidities and tobacco dependence with counseling and pharmacotherapy, patients who find smoking cessation challenging for potential undiagnosed mental health issues warrant greater attention [23]. Therefore, for persons refractory to smoking cessation clinical interventions, screening for undiagnosed mental health issues may reveal potential pathological confounders towards successful quitting smoking.

In this study, we performed mental health screenings for patients attending a tobacco treatment clinic who did not carry a diagnosis of a mental health morbidity or experience social disconnection. Our objective was to better understand the prevalence of undiagnosed mental health morbidities in persons attempting smoking cessation. We hypothesize that a significant portion of the patients attending the clinic would have an undiagnosed mental health concern whose symptoms, when present, result in ongoing significant tobacco consumption and resistance to conventional cessation approaches.

## 2. Methods

### 2.1. Study Population

All patients who attended the Tobacco Treatment and Cancer Screening Clinic (TTCSC), which has been previously described [24], between January 2021 to December 2021 were screened for potential inclusion. Screening consisted of a member or members of the TTCSC interdisciplinary team reviewing patient charts prior to their clinical appointment. If the patient had (a) no diagnosis of a mental health abnormality or disorder, (b) were not on any FDA-approved medications for mental health and (c) it was their first visit to the TTCSC, they were identified as eligible to undergo the screening. All pediatric patients (age 17 years old or younger) were excluded. Note such screening occurred through a review of the patient’s electronic medical records (EMR). The Institutional Review Board at Johns Hopkins School of Medicine approved the study, and all actions undertaken by the authors were in accordance with the Declaration of Helsinki.

### 2.2. Data Collection and Study Design

Once patients are identified prior to their appointment, they are screened on the same day of the clinical encounter prior to or after the clinical visit (within 72-h of the clinic visit). Demographic and healthcare data are collected and includes individual factors (age, race, ethnicity, cardiopulmonary disease, active oncological disease) and contextual variables (census tract linked to the patient’s respective national area deprivation index [25,26]). Smoking-related variables are also collected and include cigarettes per day, brand of cigarette and if the cigarettes contain menthol, age of starting smoking, prior cessation attempts in the last 5-years, the number of successful cessation attempts in the last 5-years (defined as off of cigarettes for more than 7-days), and the perception of the amount of current smoking during the pandemic as compared to pre-pandemic years.

Mental health and social connection screenings were conducted for all patients of the clinic who screened positive (meaning, they carried no documented mental health abnormalities or diseases in their medical charts). Specifically, mood disorders, social connection abnormalities and diseases were screened for as these tend to influence smoking behaviors and are often missed or under-reported due to factors such as insidious symptoms, clinical and cultural stigma, and implicit bias, to name a few [5,27]. First, depression and anxiety were both screened as these mental health morbidities have been associated with nicotine addiction and tobacco dependence, as well as mental health issues that are often poorly identified in traditional healthcare settings by clinicians [28,29,30]. We attempted to screen for these mood disorders using the Patient Health Questionnaire-9 (PHQ-9) (range 0–27) [31] for depression and the Generalized Anxiety Disorder-7 (GAD-7) (range 0–21) [32] for anxiety. A positive PHQ-9 was noted as a score of 10 or more, while a positive GAD-7 was also at the threshold of a 10 or more. Of note, at the end of the screening by the clinical staff member, patients were asked if they would like a referral to counseling, regardless of the score.

Patients were also screened for insight into their well-being through their social network participation as well as sense of loneliness, both of which are factors that impact smoking [33,34,35]. The abbreviated 6 item Lubben Social Network Scale (LSNS) (range 0–60) was utilized and examines the frequency and type of interactions with friends and family [36]. At risk for social isolation was a score between 0–12. For loneliness, we implemented the item UCLA Loneliness Scale (range 20–80) [37,38]. A score of 28 or more was identified as a positive finding for loneliness [38]. Of note, social network engagement and loneliness are likely to be missed or under-reported in a patient’s medical chart, and at a higher rate of being missed or under-reported as compared to depression and anxiety [27,39]. In addition to counseling, patient-centered support groups overseen by the hospital were offered to the patients at the end of the screening, regardless of outcomes.

### 2.3. Primary Outcome

The primary outcome was the incidence of screening positive for one of the aforementioned mental health or social connection morbidities.

### 2.4. Statistical Analysis

Quantitative variables were expressed as mean ± standard deviation or median (interquartile range), when appropriate, and were compared between groups of patients who screened positive for depression, anxiety, or social isolation. The comparison was performed with nonparametric Wilcoxon rank-sum test. Qualitative variables were expressed as frequency (percentage) and were compared between the aforementioned groups using Fisher exact test. When comparing patients who screened positive for a mental health abnormality, we performed group comparisons across the four mental health conditions using Kruskal-Wallis test. Factors associated with smoking consumption, specifically evaluating the outcome of “smoking more” during the pandemic were studied within mental health groups. All of the statistical analyses were executed with R statistical software version 3.2.0 and available online (r-project.org, accessed on 5 March 2022)

## 3. Results

### 3.1. Baseline Characteristics

During the 12-months of data collection in 2021, 114 new patients enrolled in the tobacco treatment clinic. Of these 114 patients, 77 (67.5%) did not have a prior diagnosis of a mental health morbidity when their medical charts were reviewed. With regards to the 37 patients who screened positive for a mental health disorder, 20 (54.1%) were reported to have depression, 11 (29.7%) anxiety, and 8 (21.6%) with other disorders (schizophrenia, schizoaffective disorder, agoraphobia, post-traumatic stress disorders). The remaining 77 patients, who did not carry a mental health abnormality, would comprise our cohort in regard to receive screening for mental health and social disconnection morbidities.

The mean age of the 77 patients was 54.3 ± 11.2 years. Fifty-two (67.5%) were females and 64 (83.1%) were Black/African American. With regards to socioeconomic status of the neighborhood in which they reside at the time of the clinic visit, the median area deprivation index (ADI) was 77 (IQR 44, 91). The mean age of smoking initiation was 19.3 ± 5.2 years. Regarding these 77 patients, all but 8 screened positive for one of the four mental health and social connection screenings. The majority of the patients screened positive on the UCLA Loneliness Scale (70 patients). Table 1 presents the patient characteristics at the time of screening, while Table 2 presents the outcomes of the mental health and social connection screenings with regards to number of patients who screened positive and the resulting scores of the 77 patients.

### 3.2. Mental Health and Social Connection Screening Findings

As mentioned, from the 77 patients who did not have a previously recorded mental health condition, upon screening we identified 71 patients who screened positive. The most common finding of a mental health concern was loneliness, with the UCLA Loneliness Scale positive in 70 (90.1%) patients, with a score of 39.4 ± 12.4 (range 22 to 74). The next most frequent positive screening occurred with the LSNS, identifying 67 (87.0%) patients at risk for social isolation. The mean score was 14.0 ± 8.3 (range 0 to 28). Anxiety, identified with the GAD-7, was identified in 59 (76.6%) patients, with a mean score of 11.3 ± 5.7. Depression, screened for with the PHQ-9, was identified in 32 (41.6%). Finally, of the 71 patients who screened positive on at least one mental health screening tool, 63 were positive on two of the screening tools, 48 on three of the screening tools, and 21 on all four of the screening tools.

### 3.3. Influence on Tobacco Dependence

For the 71 patients who screened positive, we evaluated three tobacco dependence-related variables: cigarettes per day, prior cessation attempts, and if smoking consumption increased during the pandemic. Concerning cigarettes per day, there was a statistically significant variability between the four groups (anxiety, depression, loneliness, and social isolation) (*p* < 0.001). Patients identified for loneliness by the UCLA Loneliness Scale and patients at risk for social isolation identified by the LSNS consumed more than a pack of cigarettes a day (a pack being 20 cigarettes): 27.2 ± 8.1 cigarettes per day and 24.3 ± 7.3, respectively. Patients identified with anxiety and depression smoked about a pack a day, 18.2 ± 5.3 and 18.8 ± 4.7, respectively.

With regards to smoking during the pandemic, 59 of the 71 patients said they were smoking more. Of these 59 patients, 55 screened positive by the LSNS, 50 screened positive by the UCLA Loneliness Scale, 40 by the GAD-7, and 24 by the PHQ-9. Patients who screened positive for LSNS noted consuming 10.3 ± 4.7 cigarettes per day prior to the pandemic, with current significant increase (27.2 ± 8.1) (*p* < 0.001). Patients who screened positive for loneliness identified consuming 12.4 ± 7.2 prior to the pandemic, with a current significant increase (24.3 ± 7.3) (*p* < 0.001). Patients who screened positive for anxiety and depression did not have a statistically significant change in smoking consumption. For anxiety, 16.3 ± 4.2 cigarettes per day prior to the pandemic versus 18.2 ± 5.3 (*p* = 0.531) during the pandemic. For depression, 15.3 ± 2.7 per day prior to the pandemic versus 18.8 ± 4.7 (*p* = 0.631).

With regards to cessation attempts in the prior 5-years, more patients who scored positive on the PHQ-9 identified as having attempted to quit smoking (24 of the 32 patients, 75.0%). Next were patients who scored positive on the GAD-7 (31 of the 59 patients, 52.5%), followed by patients scoring positive on the LSNS (28 of the 67 patients, 41.2%), and then patients scoring positive on the UCLA Loneliness Scale (15 of the 70 patients, 21.4%).

## 4. Discussion

In our pilot of screening patients with undiagnosed mental health issues, we found that there was a high prevalence of mental health and social connection morbidities in patients who were actively smoking and were struggling to achieve smoking cessation. The high prevalence of such mental health (anxiety, depression) and social disconnection (social isolation, loneliness) issues that are undiagnosed and thus not managed may account for fewer prior smoking cessation attempts and/or less successful outcomes. These findings also raise the possibility of potential smoking cessation achievement if and when the patient’s mental health and social connection issues are addressed and managed; thereby, the conditional response persons have to smoking when such moods and psychiatric symptoms emerge are mitigated.

The selection of the mental health and social connection morbidities of anxiety, depression, isolation, and loneliness were due to the notion that these were likely to be underdiagnosed and impact tobacco use behavior [2,34,35]. The neurotransmitter acetylcholine enhances cortical sensitivity to external stimuli, which in turn assist in increasing focus and attention [12]. Aberrant increases in acetylcholine signaling have resulted in symptoms associated with anxiety and depression [40]. Chronic elevations of acetylcholine may also result in maladaptive behaviors, such as nicotine addiction and tobacco dependence. In addition, individuals with anxiety and depression have an upregulation of high affinity nicotine acetylcholine receptors (nAChR) compared to persons without these psychiatric illnesses [41]. Thus, the tobacco usage exhibited by these patients may occur as a form of self-medication through nicotine. However, there are several studies that have emphasized once smoking cessation was achieved, there was a positive impact on psychiatric illnesses (reduced symptoms of depression and anxiety) as compared to patients who continued to smoke [42,43]. While that may be the case, the difference in our findings from these studies is that our patients were undiagnosed to begin with; therefore, to assist with the potential confounding of mood disorders such as anxiety and depression, identifying these mental health issues and navigating their impact on smoking behavior may result in higher rates of smoking cessation.

Given the high prevalence of mental health and social disconnection morbidities identified in our tobacco treatment clinic, adding mental health services as part of the clinical services for smoking cessation warrants review. In a recent systematic review, Lightfoot et al. found that psychological interventions were independently effective in reducing smoking in persons with mental health problems [44]. The authors emphasize that in their review, they were unable to identify which interventions work best for which specific patient phenotype regarding smoking cessation, and how to best implement such strategies along with smoking cessation pharmacotherapies [44]. Taking into account our findings and the high potential clinical utility from psychological interventions, exploring a dedicated tobacco treatment clinic that offers such mental health screening and therapeutics warrants consideration.

A focus on one specific mental health issue that is often challenging to manage and greatly impacts smoking cessation is loneliness. Loneliness is a negative affective state whereby a person has the perception of being isolated socially or has an insufficient quality and/or quantity of social connection [33]. Such a negative affective state has been associated with poor health behaviors, significant morbidity, and higher mortality [35,45]. Further, negative effects have been associated with smoking consumption and tobacco use relapse [35,46,47]. In our study, we found a high rate of loneliness and social isolation amongst our patients in the tobacco treatment clinic. Understanding how such affects result in tobacco use, specifically, how they induce a susceptibility to the nicotine addiction, may assist in more equitable interventions for patients struggling with smoking cessation. For instance, clinical services for tobacco dependence management may consider adding resources aimed at addressing the experience of loneliness and social isolation in identified patients, such as the utilization of support groups or tailored coaching about social connection as these efforts have shown positive outcomes in other areas where social isolation is impacting health [48,49]. These approaches could reduce, if not altogether eliminate, the negative health behavior of smoking.

Our study does have several limitations that merit attention. First, the study period occurred during the public health crisis from SARS-CoV-2 and the resulting pandemic. Therefore, it is unclear if these mental health findings were new due to the global crisis or worsened due to the crisis. Regardless of the confounding impact COVID-19 may have on our findings, these mental health issues were still prevalent and identified by patients as having a role in their inability to stop smoking. Second, our screening for mental health and social connection disorders relied solely on electronic medical record documentation, through review of morbidities and medications. It is uncertain if the patients may have had such diagnoses and were simply never documented or carried the diagnoses without the use of pharmacotherapy. Finally, these screenings were performed in individuals older than 50 years of age. It is unclear if such screenings would be useful in younger patients, such as youth and adolescents, who may be using tobacco products such as electronic cigarettes. Further investigations of the clinical utility of mental health and social connection screenings are warranted for this younger population as the electronic cigarette youth usage epidemic continues to worsen [50,51]. However, we should emphasize the feasibility of implementing such screenings, which warrants a permanent place in the clinical realm.

## 5. Conclusions

High rates of undiagnosed anxiety, depression, loneliness, and social isolation were prevalent in patients attending a dedicated tobacco treatment clinic. Such mental health concerns were identified as playing an active role in the patients’ inability to stop smoking or to resist smoking relapse; further, these morbidities appeared to influence ongoing smoking in the patients. While larger studies in more diverse populations are warranted to contextualize these results, health system investment in screening for mental health and social connection morbidities should be explored in order to provide effective tobacco dependence management as a component of high value care.

## Figures and Tables

**Table 1 ijerph-19-11755-t001:** Baseline characteristics.

	All Patients (N = 77)	Positive Screening (N = 71)	Negative Screening (N = 8)
**Age (± SD)**	54.3 ± 11.2	56.8 ± 8.8	53.1 ± 13.3
**Female (%)**	52 (67.5)	48 (67.6)	4 (50.0)
**Black/African American (%)**	64 (83.1)	60 (84.5)	4 (50.0)
**ADI (IQR)**	77 (44, 91)	81 (44, 95)	72 (63, 88)
**Age of Smoking Initiation (± SD)**	19.3 ± 5.2	14.2 ± 7.1	22.9 ± 3.2
**Brand of Cigarettes (%)**			
**Marlboro**	10 (12.9)	10 (14.1)	0 (0.0)
**Newports**	56 (72.7)	48 (67.6)	8 (100.0)
**Pall Mall**	8 (10.4)	8 (11.3)	0 (0.0)
**Other**	3 (3.9)	3 (4.2)	0 (0.0)
**Menthol Cigarettes (%)**	74 (96.1%)	71 (100.0)	3 (37.5)
**Cigarettes per day (± SD)**	22.3 ± 6.8	23.2 ± 2.4	11.3 ± 6.7
**Smoking More During the Pandemic (%)**	61 (79.2)	59 (83.1)	2 (25.0)
**Prior Quit Attempts (%)**	34 (44.2)	26 (36.6)	8 (100)
**Successful Cessation (%) ***	21 (27.2)	18 (25.4)	3 (37.5)

ADI (area deprivation index). IQR (interquartile range). SD (standard deviation). * Successful cessation is defined as 7-days without a cigarette.

**Table 2 ijerph-19-11755-t002:** Mental health screening results of the 77 patients.

Screening Tool	Screened Positive (Number, %)	Score (Mean ± SD)
GAD-7	59 (76.6)	11.3 ± 5.7
PHQ-9	32 (41.6)	15.7 ± 7.2
LSNS	67 (87.0)	14.0 ± 8.3
UCLA Loneliness Scale	70 (90.1)	39.4 ± 12.4

## Data Availability

Data is available by contacting the primary author for further discussion.

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
