# Peer review of "Prevalence of Mental Health and Social Connection among Patients Seeking Tobacco Dependence Management: A Pilot Study"

_ijerph, 2022, doi:10.3390/ijerph191811755_

Round 1

Reviewer 1 Report

Introduction:

Line 56: ‘’Persons with mental health morbidities….’’ replace the word Persons with people

It is advisable to mention about techniques for smoking cessation e.g, NRT … and their role in smoking cessation and relapse. Also

Methods:

Since the study population were attending a screening clinic including cancer, where cancer-if diagnosed in smokers- had a role in the mental illness and social isolation? Perhaps they had cancer previously?

Authors should calculate the smoking volume and try to relate it to the smoking behaviour and mental illness.

How active were the patients included in the study?

Overall:

In introduction and discussion: Aerobic exercise is a way that is being used for smoking cessation, have you considered the physical activity levels and the physiological effects the exercise imposes on patients? Enhance mood  .. etc

I recommend that authors mention a clear conclusion in relation to ‘’ do these findings are attributed to smoking?’’

Author Response

Reviewer 1:

Introduction:

Line 56: ‘’Persons with mental health morbidities….’’ replace the word Persons with people

Response: We replaced as suggested.

It is advisable to mention about techniques for smoking cessation e.g, NRT … and their role in smoking cessation and relapse. Also

Response: We have added a line suggesting smoking cessation with pharmacotherapy in the Introduction along with a reference. The sentence is:

And if diagnosed, smoking cessation in persons with mental health morbidities can be successfully achieved with counseling along with the consideration of pharmacotherapy.

Methods:

Since the study population were attending a screening clinic including cancer, where cancer-if diagnosed in smokers- had a role in the mental illness and social isolation? Perhaps they had cancer previously?

Response: In review of this particular cohort, we did not identify patients carrying a prior cancer diagnosis. Please note this is a great question, one that we will keep in mind as we move forward.

Authors should calculate the smoking volume and try to relate it to the smoking behaviour and mental illness.

Response: In our clinic, we tend to utilize smoking topography factors in an effort to understand factors that make smoking cessation challenging. Amount of cigarettes can be misleading if patients use higher end brands (that have lower burn rates) as compared to more inexpensive brands (higher burn rates, self-extinguishing at greater rates). While we have calculated cigarette consumption in our manuscript, as cigarettes per day (page 9, which is a reasonable surrogate to smoking volume), given the small cohort (71 patients) and other factors that would contribute to smoking behavior that are not immediately addressed in this manuscript (as it was not the primary target), we feel associating smoking behaviors and mental illness with smoking volume based on the data from this cohort may be misleading.

We agree this should be explored, and as our cohort grows, we will be looking to tackle this in our future studies.

Again, great recommendation, but challenging to provide a response that would not be without significant limitations and may result in misleading conclusions.

How active were the patients included in the study?

 Response: Unclear if the question is active in a sense of “actively engaging to stop smoking”. We do not gage motivation of smoking cessation, as we feel that the initiation of tobacco dependence management will be implemented and not delayed (similar to how hypertension management, or other chronic diseases, are implemented at the recommendation of a clinician; whether they are followed through warrants ongoing touchpoints). Note that all the patients did return for follow-up visits 6-weeks later, indicating a reasonable proxy that they were motivated and active.

Overall:

In introduction and discussion: Aerobic exercise is a way that is being used for smoking cessation, have you considered the physical activity levels and the physiological effects the exercise imposes on patients? Enhance mood  .. etc

 Response: Great point – we will incorporate this consideration in the future support groups that we hold. Great suggestion.

I recommend that authors mention a clear conclusion in relation to ‘’ do these findings are attributed to smoking?’’

Response: Great suggestion and we added this sentence:

Such mental health concerns were identified to play an active role in the patients’ inability to stop smoking or to resist smoking relapse; further, these morbidities appeared to influence ongoing smoking in the patients.

Reviewer 2 Report

This is an interesting and important research topic.

I am wondering why is this a pilot study instead of a full research study? 

Improvements are needed for background, literature reviews and discussion, to make it more comprehensive and clear.  

There is a need to check through the grammar, tenses, structure of sentences and the flow of the contents. 

Author Response

Reviewer 2:

This is an interesting and important research topic.

I am wondering why is this a pilot study instead of a full research study? 

Response: Great point – and it has more to do with our loss of staff and funding during the pandemic. Much of the work to start the project was complete, but with the loss of staff and budgeting, we could only pull this off as a pilot. However, moving forward, we are securing staff and will look into building from this a larger study. Therefore, this pilot and (hopeful) future publication will reaffirm the need for larger evaluations.

Improvements are needed for background, literature reviews and discussion, to make it more comprehensive and clear.  

There is a need to check through the grammar, tenses, structure of sentences and the flow of the contents. 

Response: We are taking point by point from your input. We appreciate it greatly. And we have given the manuscript a further review for grammar.

Suggest to add another short paragraph for literatures review

Response: We have added this paragraph and can be found on our tracked changes, along with 5 new references.

Suggest to include a clear objective of the study

Response: We have updated our statement to read as such:

In this study, we performed mental health screenings for patients attending a tobacco treatment clinic who did not carry a diagnosis of a mental health morbidity or experience social disconnection. Our objective was to better understand the prevalence of undiagnosed mental health morbidities in persons attempting smoking cessation. We hypothesize that a significant portion of the patients attending the clinic would have an undiagnosed mental health concern whose symptoms, when present, result in ongoing significant tobacco consumption and resistance to conventional cessation approaches.

Suggest to include references/evidences to support the suggestions.

Response: We have added two references.

As a general comment for 'discussion', I would suggest to review the flow of the presentation style, currently I find it difficult to follow through as mental health and social connections are frequently being discussed together. I am wondering if it is possible to discuss them separately based on the results? For instance, depression, anxiety, loneliness and social isolation, any differences between them?

Response: We have updated the Discussion to go from “opening paragraph”, “why we selected what we screened for”, “how to add to your clinical work-flow”, and a focus on “loneliness”. Great suggestion!

Thank you for your consideration.